# Influence of Geographical Orchard Location on the Microbiome from the Progeny of a Pecan Controlled Cross

**DOI:** 10.3390/plants12020360

**Published:** 2023-01-12

**Authors:** Kimberly Cervantes, Ciro Velasco-Cruz, L. J. Grauke, Xinwang Wang, Patrick Conner, Lenny Wells, Clive H. Bock, Cristina Pisani, Jennifer J. Randall

**Affiliations:** 1Molecular Biology and Interdisciplinary Life Sciences, New Mexico State University, Las Cruces, NM 88003, USA; 2Entomology, Plant Pathology, and Weed Science, New Mexico State University, Las Cruces, NM 88003, USA; 3USDA ARS, Southern Plains Agricultural Research Center, Pecan Breeding & Genetics, College Station, Somerville, TX 77845, USA; 4Department of Horticulture, University of Georgia-Tifton Campus, Tifton, GA 31793, USA; 5USDA ARS, Southeastern Fruit and Tree Nut Research Station, Byron, GA 31008, USA

**Keywords:** *Carya illinoinensis*, nuts, phytobiome, seeds, seedlings

## Abstract

*Carya illinoinensis* (Wangenh.) K.Koch production has expanded beyond the native distribution as the genetic diversity of the species, in part, has allowed the trees to grow under broad geographic and climatic ranges. Research in other plant species has demonstrated that the phytobiome enhances their ability to survive and thrive in specific environments and, conversely, is influenced by the prevailing environment and plant genetics, among other factors. We sought to analyze the microbiota of pecan seedlings from the controlled cross ‘Lakota’ × ‘Oaxaca’ that were made in Georgia and Texas, respectively, to determine if the maternal geographical origin influences the microbiome of the resulting progeny. No significant differences in bacterial communities were observed between the seeds obtained from the two different states (*p* = 0.081). However, seed origin did induce significant differences in leaf fungal composition (*p* = 0.012). Results suggest that, in addition to some environmental, epigenetics, or host genetic components, ecological processes, such as dispersal mechanisms of the host, differentially impact the pecan microbiome, which may have ramifications for the health of trees grown in different environments. Future studies on the role of the microbiome in plant health and productivity will aid in the development of sustainable agriculture for improved food security.

## 1. Introduction

The native range of pecan (*Carya illinoinensis* (Wangenh.) K.Koch) in North America extends from Illinois in the U.S.A south to Oaxaca in southern Mexico [1]. Pecans grow under diverse climatic conditions. Climates in the native areas are humid with mean summer temperatures as high as 27 °C with extremes of 41 to 46 °C. Mean winter temperatures vary from 10 to −1 °C, with extremes of −18 to −29 °C [2]. In addition to the broad climatic spectrum, pecans can grow in a variety of different soil types, including acidic, alkaline, loamy, moist, rich, sandy, well-drained, wet, and clay soils [3].

The commercialization and cultivation of pecan is relatively recent; with grafted orchards first established at the end of the nineteenth century, and formalized pecan breeding beginning in 1931 [4]. Pecan breeding programs have focused on increasing pecan nutmeat size and quality, and conferring disease resistance [5]. Among the improved cultivars, ‘Lakota’ was released in 2007 by the United States Department of Agriculture (USDA) and the Kansas Agricultural Experiment Station due to its high nut quality, yield potential, scab and aphid resistance, and overall tree strength [6]. ‘Lakota’ originated from a 1964 controlled cross between ‘Mahan’ and ‘Major’ performed by L. D. Romberg in Brownwood, Texas [6]. As a result of this cross, ‘Lakota’ has the larger nut size and thin shell of ‘Mahan’ and the cold tolerance and scab resistance of ‘Major’. Recent genetic analyses have identified genes within the 1.41 Mb region of the ‘Lakota’ genome with homologs found in other species known to be involved in the acquisition of nutrients, plant development, and defense responses. In addition, 46 orthogroups were identified within the pan-genome database of the 1.41 Mb region, of which eight were specific to ‘Lakota’ [5]. ‘Oaxaca’ (87MX3-2.11) is a pecan tree that originated from an open pollinated seed collected in 1987 from a putatively native pecan tree, with a tree trunk diameter of 165 cm, growing south of Oaxaca, Mexico at the southern extent of pecan’s native range [5,7]. ‘Oaxaca’ was found to have the least heterogeneity of any pecan accessions studied to date [8]. Due to the low level of heterozygosity compared to other pecan genotypes, it was selected to develop a chromosome level reference genome for pecan [5]. Furthermore, pollen from ‘Oaxaca’ was used to make a controlled cross with ‘Lakota’ trees from multiple orchards in Georgia and Texas. The cross was made for the development of shared mapping populations in the two states [5]. Scab, a fungal disease caused by the plant pathogenic fungus *Venturia effuse* (G. Winter) Rossman & W.C. Allen, impacts pecan production in humid environments such as Georgia and Texas [9,10,11]. Seedlings from the ‘Lakota’ × ‘Oaxaca’ crosses are under evaluation for the phenotypic expression of a number of traits, including their response to scab [5]. Analyses are ongoing to compare phenotypic scab severity between seedlings from geographical locations of the Lakota mother trees.

Within the U.S.A., Georgia and Texas are among the top four states in annual pecan nut production [12]. There are six distinct soil provinces within the state of Georgia, and the Tift County soil series supports 27% of the state’s farmland [13]. The mean high and low temperatures for Tift County, Georgia are 25 °C and 12 °C, respectively, with an average annual precipitation of 120.1 cm [14]. Texas, one of the largest states in the U.S.A., has more than 1300 different soil types and is considered to have twenty-one major land resource areas that are similar in soil, vegetation, and climate [15]. The preeminent soils in Burleson County, where the mapping population is located, range from heavy clay to silt loam [16]. The mean annual precipitation in Burleson County is 1016 mm with an annual average high and low temperature of 34 °C and 12 °C, respectively [17]. Brown County is an arid location with cold winters and deep soils [18]. The mean annual precipitation in Brown County is 773 mm with an annual average high and low temperatures of 26 °C and 11 °C, respectively [19]. Although Texas contains several Köppen climate classification types, both Tift County, GA and Burleson County, TX are classified as having a humid subtropical climate [20,21]. The temperature and rainfall, and associated humidities in Burleson County in Texas and Tift County in Georgia during the growing seasons are conducive to various plant diseases, and in particular pecan scab. But despite the climate similarities between the locations, some diseases differ in prevalence between the two locations. Besides scab, Georgia pecans are sporadically affected by leaf and nut anthracnose (*Colletotrichum acutatum* J.H. Simmonds), powdery mildew (*Microsphaera penicillate* (Wallr.) Lév.), phytophthora shuck rot (*Phytophthora cactorum* (Lebert & Cohn) J. Schröt.), and zonate leaf spot (*Cristulariella moricola* I. Hino) [22]. Conversely, pecan trees in Texas are sporadically affected by downy spot (*Mycosphaerella caryigena* Demaree & Cole) and fungal twig die back (*Phomopsis* spp. P.A. Saccardo & C. Roumeguère and *Botryosphaeria* spp. V. de Cesati & G. de Notaris), to name a few [13].

Nonetheless, a plant’s ability to thrive under any given condition is not solely dependent on its inherit genetics, but also its phytobiome; the combination of the plant, the microbial communities associated with it, and the prevailing environmental conditions [23]. Studies of bacterial and fungal communities in plants have revealed the influence that host genotype and age, environment, and plant organ have on microbial community composition [24]. Plant genotype-specific processes have been developed to mediate microbiome assembly [25]. Moreover, the microbiome assembly of specific plant genotypes was observed to change throughout the development of the plant. For example, specific signals in the rhizosphere microbiome of sorghum were reported to be detectable only during the later developmental stages of the plant [26]. Although studies have identified the factors influencing microbial composition in plants, other studies working towards unraveling the extent to which each factor plays a role in the assembly of microbial communities are gaining traction. Studies on *Brassica napus* C. Linnaeus seeds have identified environment to have a stronger effect on the microbiome composition than the host genotype [27]. The microbiota present in different plant organs (i.e., leaves, stems, roots), however, do differ and are shaped by various factors [28,29,30].

Efforts to improve crop production and ensure food security, due to a growing global population and climate change, are paramount. Understanding the microbiota present in pecan will allow us to better understand adaptation, selection and dispersal of both deleterious and beneficial associates, positively impacting cultural systems for improved breeding practices. As information on the microbiome of pecan is new and limited, the impact of genotype or environment is unknown. In this study, we sought to identify the microbial composition of the controlled pecan cross ‘Lakota’ × ‘Oaxaca’. The maternal trees (‘Lakota’) were located in either Georgia or Texas, and the pollen (paternal) from ‘Oaxaca’ was collected in Georgia. Our objective was to gain insight into the influence that maternal tree origin has on the composition of the microbiota associated with developing pecan seedlings.

## 2. Results

### 2.1. Seed Characteristics

The seeds obtained from the controlled crosses were measured for various characteristics, including length, width, height, weight, volume, and density, prior to planting. Results from these measurements are shown in Table 1. The seeds from each individual maternal accession were similar with no statistical differences.

### 2.2. Microbial Diversity

Microbiome sequence analysis of the bacterial communities associated with the pecan seedlings resulted in 919 operational taxonomic units (OTU) with the exclusion of mitochondria, chloroplast, unknown, and ambiguous taxa. Of the 919 bacterial OTUs, 373 were identified from the Texas seedlings, while 572 were identified from the Georgia seedlings. Analysis indicated that 26 OTUs were shared between the two states. A total of 264 bacterial OTUs aggregated by taxonomic family were identified and analysis revealed 81 families were shared between the two states. In the Texas seedlings, 47 unique OTUs aggregated by taxonomic family were identified, while 136 were identified in the Georgia seedlings. Of the 17,635 fungal OTUs analyzed, 8975 were identified in the Texas seedlings while 8664 were identified in the Georgia seedlings, with only 4 OTUs shared by seedlings between the two source states. A total of 30 fungal OTUs aggregated by taxonomic family were identified of which 10 fungal families were shared between the two states. In the Texas samples, 12 unique OTUs aggregated by taxonomic family were identified while only eight were identified in the Georgia samples. Furthermore, fungal analysis revealed that of the 17,635 OTUs, 17,513 were unknowns.

The most abundant bacterial families associated with seedlings from seed at both locations were the *Burkholderiaceae* G.M. Garrity, *Enterobacteriaceae* Rahn, *Anaerolineaceae* T. Yamada, *Rhodocyclaceae* G.M. Garrity, *Paenibacillaceae* G.M. Garrity, *Flavobacteriaceae* H. Reichenbach, *Steroidobacteraceae* Q. Liu, *Moraxellaceae* R. Rossau, *Methanoregulaceae* S. Sakai, and *Propionibacteriaceae* E.A. Delwiche (Figure 1A). Among fungal families, ITS sequence analysis revealed *Aspergillaceae* Link, *Pseudeurotiaceae* D. Malloch & R.F. Cain, *Malasseziaceae* Denchev & R.T. Moore, *Pleosporaceae* Nitschke, *Sporidiobolaceae* R.T. Moore, and *Nectriaceae* L.R. & C. Tulasne to be the most abundant in the ‘Lakota’ × ‘Oaxaca’ seedlings (Figure 1B). Of the bacterial families identified in both states, *Burkholderiaceae* was found to be relatively the most abundant in seedlings. The fungal family *Aspergillaceae* was identified in seedlings grown from seed collected from both states with Georgia sourced seedlings having the highest relative abundance. Conversely, the fungal classes *Pseudeurotiaceae* and *Malasseziaceae* were identified in both states, but Texas had relatively the greatest abundance of both families. The OTUs from the shared families were used for functional inference and the results for 16S and ITS can be found in Appendix A. The results using the Enzyme Commission (EC) database identified six functional enzymatic categories (oxidoreductases, transferases, hydrolases, lyases, isomerases, and ligases) for both bacterial and fungal families. The EC numbers divided into their sub-subclasses to the identified inferred enzyme can be found in Appendix A.

### 2.3. Alpha Diversity

Phylogenetic alpha diversity analysis of the bacterial communities identified in the ‘Lakota’ × ‘Oaxaca’ seedlings revealed no statistical differences between the seedlings from seed of the parent trees in Georgia when compared to those seedlings from seed of the parent tree in Texas (Figure 2A). Phylogenetic alpha diversity analysis of the fungal communities identified followed a similar pattern to those of the bacterial communities (Figure 2B). The individual seedling samples from each state, respectively, indicate a homogenic relationship for both bacterial and fungal analyses. Based on the Mann–Whitney and Kruskal–Wallis analyses for both the bacterial and fungal diversities associated with the pecan seedlings from the Texas and Georgia crosses, *p* values indicated a lack of evidence for differences in alpha diversity (bacterial *p* = 0.5 and fungal *p* = 0.6).

### 2.4. Beta Diversity

Beta diversity analysis of the bacterial communities from the ‘Lakota’ × ‘Oaxaca’ seedlings from parent trees in Georgia and Texas, indicated a lack of similarity and were dispersed throughout the principal coordinate analysis plot (Figure 3A). Conversely, beta diversity analysis of the fungal communities resulted in a different outcome (Figure 3B). The samples showed that communities from seedlings that originated from Georgia clustered tightly together, while communities from only trees of the seedlings that originated from Texas clustered together. An Adonis test comparing the seedlings from parent trees in Georgia and Texas revealed that the clusters were statistically significant only for the fungal communities. Bray–Curtis was used as measure for beta diversity when conducting Adonis tests of bacterial and fungal communities. The analyses resulted in R^2^ values of 0.1264 (bacterial) and 0.9413 (fungal) with *p* values of 0.081 (bacterial) and 0.012 (fungal), respectively (Figure 3A,B). Heat map analysis further demonstrated unique fungal communities present in the Georgia and Texas seedlings (Figure 3D). However, of the 25 most different OTUs across all samples, all 25 were unidentified fungal families (Appendix A). To discern differences between the OTUs identified, the unidentified OTUs were removed (Appendix A, Figure 3D).

## 3. Discussion

Few differences were observed in the occurrence, diversity and relative abundance of the bacterial families on seedlings of the ‘Lakota’ × ‘Oaxaca’ cross from Texas and Georgia. However, there were several differences in the occurrence, diversity and relative abundance of the families of fungi. Despite one of the Georgia orchards receiving fungicide treatments, differences in beta diversity between the Georgia samples were not observed as the Georgia seedlings clustered tightly together (Figure 3B). Based on these observations, we contend that geographical origin of the pecan seed in our study may play a role in the leaf microbial composition of the resulting seedlings. The trees from the different geographical locations may have had variable nutrient compositions (this was not determined or measured) and could account for differences in the resulting seed/seedling microbiome. Furthermore, the possibility of epigenetic differences between the mother trees cannot be discounted. Though surface disinfection may not eliminate all microorganisms from the shell surface, it is important to note that the seeds from both Georgia and Texas underwent surface sterilization and were planted in the same potting mix and grown together in a contained greenhouse with a controlled environment.

Fungicide treatment of pecan trees in orchards is a common practice where diseases such as scab negatively impact crop production. Seed-borne pathogens in other crops [30] are a common occurrence that negatively impact crop production. Since antibiotics are known to shift microbial communities in pecan [31], the effects that fungicide treatments may have on the microbial composition of maternal trees, and thus seed-to-seedlings, should be considered. Research on wheat (*Triticum aestivum* C. Linnaeus) leaves and common bean (*Phaseolus vulgaris* C. Linnaeus) leaves revealed a shift in relative fungal abundance of epiphytic and endophytic fungi after fungicide treatments [32,33,34]. As reported in the literature, transmission of microbial organisms from seed to seedling is a natural occurrence [30,35,36,37]. Thus, if a shift in microbial composition of the maternal plant occurs, the microbiome of the resulting seeds may in turn be altered due to the influence that maternal plants have in shaping seed microbiomes [38]. Although the effects of fungicide treatment on seed microbiota are less well documented, a similar disruption of the seed microbiome and vertical transmission from seed-to-seedling may occur [39,40,41,42,43]. Based on these reports, it is reasonable to consider that fungicide treatments may not only affect the maternal plant microbiome but also their respective progeny and related benefits to seedling growth and establishment.

Microbial communities have been observed to be influenced by the local environment as reported with *Arabidopsis thaliana* G. Heynhold grown at different sites in Europe [44]. Thus, although some knowledge exists on the effect of location on the plant microbiome, there is a gap in knowledge regarding the influence of seed source geographical origin on the composition of microbial communities associated with the resulting seedlings. Our study only identified microbial communities present in leaf tissues of seedlings grown from seeds obtained from the same cross in different locations, but we were able to gain some insight into the impact that geography plays on tree microbial composition. Although we did not narrow down our analysis to species, it is an important observation that the family *Xanthomonadaceae* G.S. Saddler & J.F. Bradbury was identified in GA and TX. The species *Xylella fastidiosa* J.M. Wells, the causative agent of pecan bacterial leaf scorch belongs to the family *Xanthomonadaceae*. *Xylella fastidiosa* has been shown to be endemic in GA orchards [45] and most recently seed to seedling transmitted in pecan [30]. Functional inference analysis of sequences from the families referenced in this study (Appendix A) identified EC numbers that may be involved in the disease suppression of plants. For instance, EC: 3.2.1.14 was identified in the bacterial microbiomes of the seedlings which corresponds to chitinases. Studies have found that some bacterial chitinases may serve as potential biological control agents against phytopathogenic fungi known to cause a variety of plant diseases [46,47]. Furthermore, EC number 1.11.1.7 was also identified in both bacterial and fungal microbial communities corresponding to peroxidases; often associated with defense mechanisms against plant pathogens (Appendix A) [48].

The lack of differences observed in the occurrence or abundance of components of bacterial communities between our Georgia and Texas samples was unexpected, but similar results have been reported in other plant microbiome studies. The bacterial and fungal communities of strawberry plants located at North American and European sites were examined and found leaf bacterial populations were not as unique between sites compared to the fungal communities present in leaves [49]. Our findings corroborate those of Mittelstrass et al. [49] and suggest that ecological processes, such as dispersal mechanisms of the host, differentially affect bacterial and fungal communities.

In a study on the microbiome of cottonwood trees (*Populus tremula* C. Linnaeus), leaves were found to have the least diversity and abundance of microbes [28]. Therefore, in addition to the influence that ecological processes have on microbial composition, plant tissue type has an influential role on the associated microbiota. Furthermore, host genotype was discovered to play an important role in shaping plant microbiota, including that associated with leaves, stems, and roots of 56 tree species [50]. While our study revealed significant differences in fungal communities associated with pecan seedlings from seeds originating in Georgia or Texas, the majority of the fungal sequences were unidentified. Thus, the identification and correlation of specific fungi to either Georgia or Texas remains unknown. The identification of key differences in the environmental and climatic conditions between Tift County, Georgia and Burleson County, Texas, is a starting point for understanding the relevance of our findings.

Based on the Köppen climate classifications, Tift County, GA and Burleson County, TX share a similar climate of hot humid summers and cool winters but vary widely in soil type. Thus, the fungal differences observed in the seedlings, originating from the different states, presents the possibility of microbial carry over from locally existing microbiota influenced by environmental factors as reported in the study by Mittelstrass et al. [49]. Bacterial and fungal communities on strawberry were affected by the Ca/Mg ratio and other factors in North America, while in Europe, factors, including evapotranspiration, soil clay content, and copper, were found to be predictors of bacterial and fungal richness. Furthermore, leaf microbiome diversity was associated with Ca and K availability in North America while in Europe, Ca and K availability were highly variable and found to have no major effect. Environment-by-environment (E × E) interactions associated with community richness were identified in the European samples where the availability of K was outweighed by a crossover interaction between K and lime, water vapor pressure, and pH in the rhizosphere. Nutrient availability is inevitably a driving factor in plant microbial composition. Thus, the nutrient uptake that the seed experienced while on the tree before harvest could possibly explain the differences we observed in the microbiome analyses of pecan seedling leaf tissue from two states; as reported in the literature [51,52,53], different microbes require different nutrients which may also be a driving factor for microbial selection in plants. Interestingly, one of the Texas samples, the only seedling used in the study from Brown County, appeared to be an outlier in the beta diversity analysis of fungal communities (Figure 3B). Although we do not have enough samples from Brown County to make any statistical analyses, this observation does indicate that future research should focus on determining if differences between counties or local environments influence microbial composition. Research in soybean (*Glycine max* E.D. Merrill) found that the microbiome was modulated by the seed rather than the soil microbiome, even after disruption through surface sterilization and irradiation [54]. This, along with previous literature, indicates that the primary source for the microbiota in the plant environment comprises the microbial communities present in the seed [54,55,56,57].

Further research is needed to fully understand the extent to which different environmental and host factors influence pecan plant microbial composition. Although microbial studies in pecan are in the early stages, research seeking to understand the influence that pecan maternal plants have on the resulting microbiome of pecan seedlings was recently published [58]. In this study, differences in microbial composition (bacterial and fungal) between seedlings from different maternal pecan cultivars were observed and a core microbiome for pecan was reported. Interestingly, no differences were reported between the seedlings (from open pollination) within each respective maternal cultivar, proposing a signature microbiome. The core families identified by Cervantes et al. [58] were also identified in the ‘Lakota’ × ‘Oaxaca’ seedlings of this study. Results of studies to better understand the role of the environment, the host genotype, and maternal factors will eventually contribute to understanding the role of the microbiome in pecan tree health in contrasting environments. As microbiome studies continue on pecan and other plant species, the role of the microbiome in plant health and productivity will aid in the development of sustainable agriculture for improved food security.

## 4. Materials and Methods

### 4.1. Seed Source and Collection of Pecan Seedling Material

Pollen of ‘Oaxaca’ was collected in 2017 from the ortet located at the USDA-Southeastern Fruit and Tree Nut Research Station, Byron, Georgia and stored at −20 °C until use. Multiple ‘Lakota’ trees growing in two separate orchards located in Tift County, Georgia and two counties in Texas (Brown and Burleson counties) were pollinated with pollen from ‘Oaxaca’ resulting in the controlled cross ‘Lakota’ × ‘Oaxaca’ [5]. As part of management protocols, the orchards with the ‘Lakota’ mother trees in Texas were treated with fertilizers and herbicides at the time of the experiment in 2017; however, no fungicide treatments were applied. Between April and August of 2017, one orchard in Georgia received three herbicide treatments, two fertilizer treatments, and six fungicide treatments, while the other orchard in Georgia received three herbicide treatments, one fertilizer treatment, and no fungicides. Seeds from the resulting cross were collected and measurements of the seeds were recorded (Table 1; [59]). For this study, a total of 10 seeds (5 seeds from each state) were stratified prior to being shipped to New Mexico State University (NMSU, Las Cruces, New Mexico) where they were surface sterilized: rinsed in soapy water for 30 s, rinsed with 70% Ethanol, two washes with a 20% sodium hypochlorite solution for 30 min and rinsed with deionized water. Seeds were planted in hydrogen peroxide treated Lamberts Potting Soil Mix (Lambert, Québec, Canada) in 10-cm square pots and grown in a quarantine greenhouse at NMSU, at a temperature of 28–35 °C. Leaves from the seedlings (with an average height from the base of the plant of 15.24 cm; Appendix A) were collected and stored in a −20 °C freezer until further processing.

### 4.2. Genomic DNA Extraction

Leaf tissue was ground in liquid nitrogen with a mortar and pestle and transferred to microcentrifuge tubes. Samples were stored at −20 °C until DNA extraction. Total gDNA was isolated using a Qiagen DNeasy Plant Mini Kit (#69106, Qiagen, Hilden, Germany) according to the manufacturer’s protocol with the exception of the elution step, in which we eluted in 50 μL of Tris-EDTA (TE) buffer. The concentration and quality of the extracted DNA were measured using an IMPLEN NanoPhotometer P-Class 360 spectrophotometer (Westlake Village, CA, USA). DNA was stored at −20 °C to prevent degradation.

### 4.3. Next Generation Sequencing and Microbiome Analyses

Library preparation and next generation sequencing of the DNA samples (using a concentration between 21 ng/μL to 66 ng/μL per sample) were performed by the Genomics Center at the University of Minnesota (Minneapolis, Minnesota, USA) using an Illumina MiSeq platform with a maximum read length of 2 × 300. To identify and quantify bacteria and fungi in the samples, sequences of the V4 and V6 domains of the 16S ribosomal (r) RNA l, and sequences of the internal transcribed spacer (ITS1) region were used, respectively. Sequence pairing, trimming, quality control, operational taxonomic unit (OTU) clustering, chimera filtering, further clustering, alpha and beta diversity analysis, and statistical analysis of the sequence results were performed using the Microbial Genomics Module version 21.1 of CLC Genomics Workbench 21.0.4 (Qiagen). Further statistical analyses were performed in R [60]. The parameters used were those configured by the workflow of the Microbial Genomics Module with some modifications. Paired-end reads had a minimum and maximum distance of 100 and 550 bp, respectively, and failed reads were discarded. Read trimming included quality and adapter trimming, and sequence filtering. Reads were trimmed from the 3′-end to a fixed length of 210 bp. Samples were filtered based on a minimum number of reads of 50 with a minimum percent from the median of 25. The SILVA 16S v132 bacterial database https://www.arb-silva.de/documentation/release-132/ (accessed on 10 April 2021) and the full UNITE v7.2 https://unite.ut.ee/ (accessed on 11 April 2021) eukaryotic database were used as references for OTU clustering and identification of bacterial and fungal communities, respectively. Clustering threshold was set to 97% similarity. Settings included the creation of new OTUs with a taxonomy similarity percentage of 80, minimum occurrences of 1, fuzzy match duplicates, and find best match were also used. Furthermore, chimeric sequences were removed, and mitochondrial and chloroplast sequences were filtered for 16S analysis. For OTU clustering, visualization of the fungal communities and unknowns were removed.

The Microbial Genomics Module MUSCLE tool was used to construct a phylogenetic tree using a maximum likelihood method with the Jukes Cantor nucleotide substitution model, based on a multiple sequence alignment (MSA) of the OTU sequences. The resulting phylogenetic trees were used for alpha and beta diversity analyses. A maximum sampling depth of 5000 reads was used for rarefaction analysis of bacterial and fungal communities. Box plot visualization of alpha diversity was generated using CLC Microbial Genomics Module version 21.1. The non-parametric Mann–Whitney and Kruskal–Wallis tests were used for alpha diversity comparisons. For beta diversity and principal coordinate (PCo) analysis, the Bray–Curtis dissimilarity metric was used along with the cmdscale function in R. To quantify the influence that the state from which seeds were collected has on the microbial communities of the resulting seedlings, the Adonis test was performed using the adonis2 function in the vegan library of R. The Bray–Curtis dissimilarity metric was used as input for the adonis2 function, which was computed using the vegdist function, also available in the vegan library. Two summary statistics to assess the importance of the sources of variation (in the analysis of variance table) were computed in the adonis2 function: the R^2^ value (computed as the ratio of the Sum of Squares of the factors and the Total Sum of Squares) and *p* value. Hierarchical clustering heat maps were generated using the 25 most different OTU’s across all samples (FDR *p* = 0) with the CLC Microbial Genomics Module version 21.1. Inferred functions of OTUs respective to the microbial families shared between the two states were conducted using PICRUSt2 and the CLC Microbial Genomics Module Infer Functional Profile tool. The EC functional term counts associated with 16S regions in prokaryotes and the ITS regions of fungi and the EC databases were used as parameters. All 16S and ITS sequence data were deposited at the National Center for Biotechnology Information (NCBI, Bethesda, MD, USA) Sequence Read Archive as part of BioProjects PRJNA803499 and PRJNA803511, respectively.

## 5. Conclusions

This study provides insights into the influence that maternal geographical location has on the microbiome of the resulting progeny. Results revealed that seed origin significantly impacted the fungal microbial communities that were present on the seedling leaf tissues as opposed to those of bacterial communities. Leaf fungal communities of seedlings from the maternal trees located in Georgia were significantly different than the seedlings from the maternal trees located in Texas (*p* = 0.012). Although one of the Georgia orchards received fungicide treatments, no differences in fungal beta diversity were observed between the seedlings whose seeds originated from Georgia as these samples tightly clustered together on the PCoA analysis. No significant differences in bacterial communities were observed between the seeds obtained from the two different states (*p* = 0.081). Though differences were not observed in the occurrence nor abundance of bacterial communities between our Georgia and Texas samples, similar results have been reported in other plant microbiome studies. The elucidation of the role played by the microbiome in the health and productivity of pecans and other plant species will aid in the development of sustainable agriculture for improved food security.

## Figures and Tables

**Figure 1 plants-12-00360-f001:**
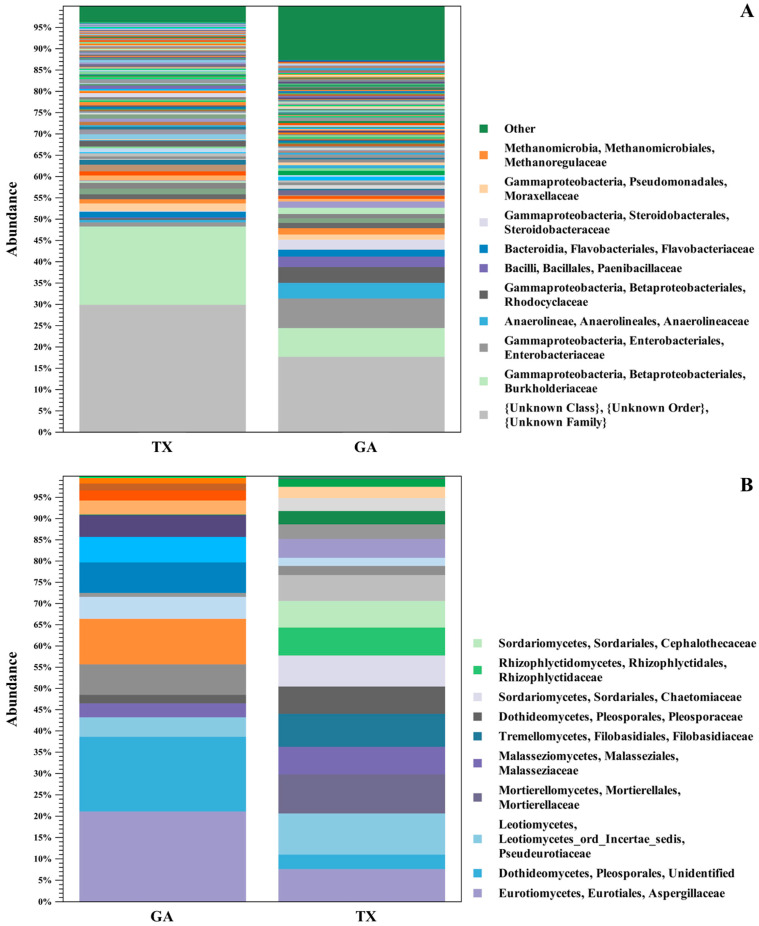
16S and ITS OTU Clustering. Relative abundance of bacterial (**A**) and fungal (**B**) populations in seedlings of pecan grown from seed of the same controlled ‘Lakota’ × ‘Oaxaca’ cross from trees in Texas (TX) and Georgia (GA). The stacked bar chart shows OTU clustering at the level of taxonomic family of the bacterial and fungal microbial communities associated with the seedlings from the Georgia and Texas seed samples.

**Figure 2 plants-12-00360-f002:**
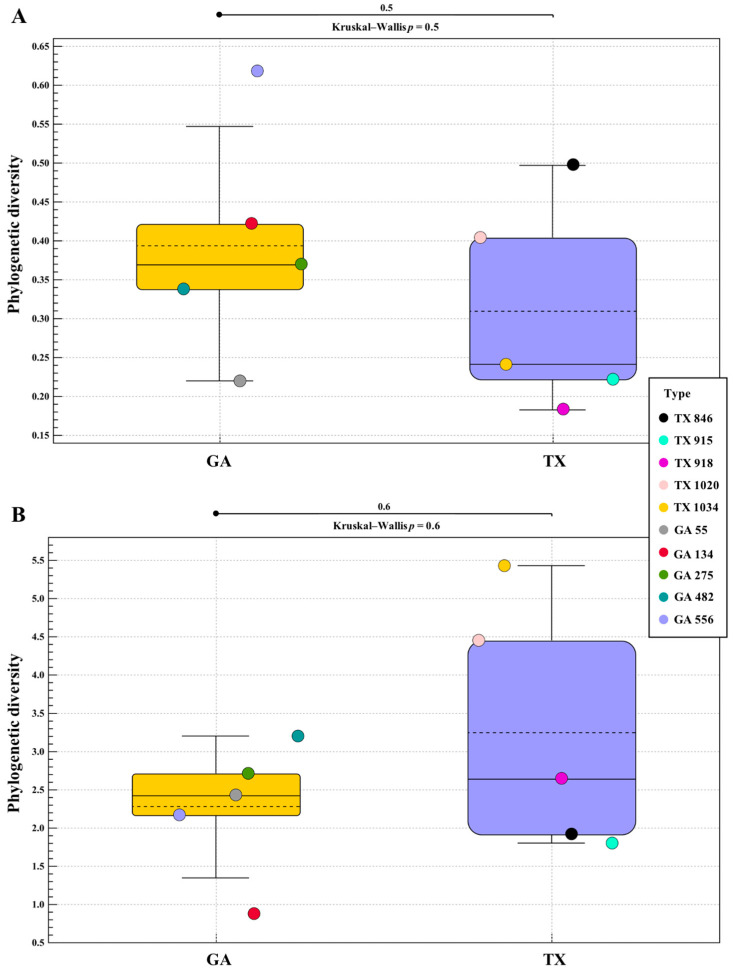
16S and ITS Phylogenetic Alpha Diversity. Phylogenetic alpha diversity of bacterial (**A**) and fungal (**B**) populations in seedlings of pecan grown from seed of the same controlled ‘Lakota’ × ‘Oaxaca’ cross from trees in Texas (TX) and Georgia (GA).

**Figure 3 plants-12-00360-f003:**
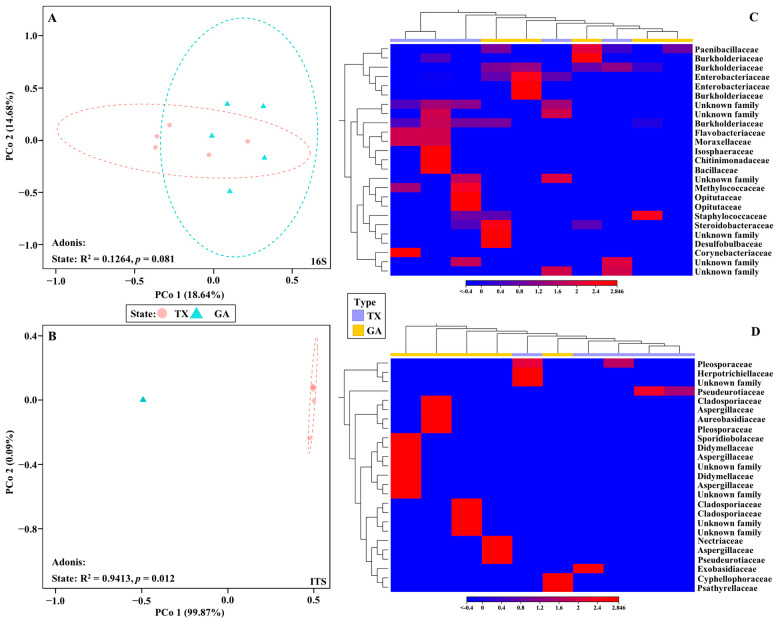
16S and ITS Beta Diversity. PCoA visualization of bacterial (**A**) and fungal (**B**) populations in seedlings of pecan grown from seeds of the same controlled ‘Lakota’ × ‘Oaxaca’ cross from trees in Texas (TX) and Georgia (GA). The colored shapes represent samples from their respective state (red circle represents Texas and the blue triangle represents Georgia). Heat map visualization of bacterial (**C**) and fungal (**D**) beta diversity in seedlings grown from seeds of the same controlled ‘Lakota’ × ‘Oaxaca’ cross from trees in Texas (TX) and Georgia (GA). Here we show the 25 most abundant OTUs with their taxonomic assignment at the family level.

**Table 1 plants-12-00360-t001:** ‘Lakota’ × ‘Oaxaca’ seed information and measurements. The seeds used in this study obtained from the controlled ‘Lakota’ × ‘Oaxaca’ crosses were individually measured.

State	Year Crossed	Maternal	Pollen	Stratified	Maternal Accession	Seed Number	Nut					
							Length (mm) ^1^	Width (mm) ^2^	Height (mm) ^3^	Weight (g)	Volume (g) ^4^	Density (g) ^5^
Georgia (GA)	2017	‘Lakota’	‘Oaxaca’	Dec 2017	Lenny1	55	41.71	23.3	20.74	7.76	9.71	0.8
					Lenny1	134	40.61	23.55	20.84	7.36	9.22	0.8
					Lenny2	275	41.39	23.89	22.18	8.59	10.7	0.8
					PVG 14-7	556	44.72	23.94	21.76	7.48	11.45	0.65
					PVG 14-7	482	46.98	24.35	21.67	5.93	11.69	0.51
Texas (TX)	2017	‘Lakota’	‘Oaxaca’	Dec 2017	BRW 153-38	846	42.76	20.92	21.43	7.45	9.27	0.8
					CSD 7-4	918	35.38	22.83	22.41	7.63	9.53	0.8
					CSD 7-4	915	37.94	23.08	21.37	7.38	9.24	0.8
					CSD 14-8	1034	36.17	22.94	22.74	7.67	9.54	0.8
					CSD 14-8	1020	38.38	20.44	20.39	5.21	8.11	0.64

^1^ Nut length is measured from apex to base. ^2^ Nut width is measured across the plane of the suture at the widest point. ^3^ Nut height is measured perpendicular to the plane of the suture at the widest point. ^4^ Nut volume is determined based on the nut’s buoyancy. ^5^ Nut density is determined by dividing the nut weight by the nut volume.

## Data Availability

Publicly available datasets were analyzed in this study. This data can be found here: [https://www.ncbi.nlm.nih.gov/bioproject/PRJNA803499] (accessed on 22 December 2022) and [https://www.ncbi.nlm.nih.gov/bioproject/PRJNA803511] (accessed on 22 December 2022).

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
