# Peer review of "Influence of Geographical Orchard Location on the Microbiome from the Progeny of a Pecan Controlled Cross"

_plants, 2023, doi:10.3390/plants12020360_

Round 1
Reviewer 1 Report
Manuscript Influence of Geographical Orchard Location on the Microbiome from the Progeny of a Pecan Controlled Cross by Kimberly Cervantes , Ciro Velasco-Cruz , LJ Grauke , Xinwang Wang , Patrick Conner , Lenny Wells , Clive H Bock , Cristina Pisani , Jennifer J Randall reviews the results research on the influence of genetic origin on the microbiome using the pecan as an example.
The manuscript contains an extensive review of the literature on the culture itself and current research in this rapidly growing area of research. In my opinion, this part is redundant and can be moved to the discussion. Other parts of the manuscript correspond to the requirements, except that the authors forgot to formulate a conclusion, and place it in the appropriate section. Another technically solvable problem is falling letters in the designation under the line in Figure 1 and too small letters of the names in Figure 3.
In addition, this study raises some questions from the point of view of the methodological approach. The authors believe that the plant genome is responsible for the interaction with bacteria, fungi, and actinomycetes, and from this position they build a research strategy and discuss the results. Indeed, the relationship between the genome and the composition of microorganisms has been previously established. However, it should not be forgotten that epigenetic regulation can play an important role in these processes, which is also preserved in the generation and can be different depending on the factors that influenced the mother plant. It was probably difficult to take this into account in the experiment, since it was necessary to use at least seeds collected in different years, which was not done. If this has not been done, this does not mean that it is possible to avoid this problem in the discussion. This is worth adding as a source to be taken into account in the future. On the other hand, it is surprising that the authors do not suggest that surface sterilization cannot eliminate all microorganisms or spores, and appropriate controls were not provided. Tenicheski not clear question and statistics and cultivation. Have all the seeds sprouted (which is doubtful, and given that there are only 5 of them already, it is fraught with errors). It is also not clear whether these seeds were of equal value and whether the developing seedlings were the same, which is also doubtful. For further evaluation, it would not be bad to add the characteristics of the plants themselves, and a photo, the size of above-ground organs and roots, dry / wet weight. Separately, it is not obvious how many repetitions the authors used and whether the indicators differed in two independent plants from the progeny of the same tree, and how this looks from the standpoint of statistics and the admissibility of conclusions.
Unfortunately, without the elimination of these remarks, the work will remain unreliable and will be controversial. I recommend the authors to review these problems and eliminate them.
Author Response
Thank you for your time and effort reviewing our manuscript. We have tried to address your edits and concerns.
- “except that the authors forgot to formulate a conclusion, and place it in the appropriate section.”
Response: We have addressed the conclusion section and fixed other issues with formatting such as the references.
- “Another technically solvable problem is falling letters in the designation under the line in Figure 1 and too small letters of the names in Figure 3.”
Response: We have addressed the font size of the letters in Figure 1 and Figure 3. We ensured that the figures met the font guidelines in Plants.
- “However, it should not be forgotten that epigenetic regulation can play an important role in these processes, which is also preserved in the generation and can be different depending on the factors that influenced the mother plant. . It was probably difficult to take this into account in the experiment, since it was necessary to use at least seeds collected in different years, which was not done. If this has not been done, this does not mean that it is possible to avoid this problem in the discussion. This is worth adding as a source to be taken into account in the future.”
Response: We agree with the reviewers and thank them for this suggestion. We have added the possibility of epigenetic regulation to the abstract (line 24) and to the discussion section (lines 224-225). This manuscript only has seeds from a 2017 cross that were germinated in a quarantine GH facility. We do have research plans in motion to address the possibility of epigenetic effects on controlled crosses with the same mother plants in multiple years.
- “On the other hand, it is surprising that the authors do not suggest that surface sterilization cannot eliminate all microorganisms or spores, and appropriate controls were not provided. Tenicheski not clear question and statistics and cultivation.”
Response: There was no intention to suggest that surface disinfection can eliminate all microorganisms. We have added a comment in lines 225-228 that surface disinfection may not eliminate all microrganisms from the outside of the shell.
- “Have all the seeds sprouted (which is doubtful, and given that there are only 5 of them already, it is fraught with errors). It is also not clear whether these seeds were of equal value and whether the developing seedlings were the same, which is also doubtful. For further evaluation, it would not be bad to add the characteristics of the plants themselves, and a photo, the size of above-ground organs and roots, dry / wet weight. Separately, it is not obvious how many repetitions the authors used and whether the indicators differed in two independent plants from the progeny of the same tree, and how this looks from the standpoint of statistics and the admissibility of conclusions.”
Response: The reviewer is correct that we started with many more seeds than the five from each location. The germination rate was less than 50% so we used the seeds that germinated for the study. The seed characteristics were measured, and these measures are provided in Table 1 for the seeds used in this study (lines 120-130). Table 1 also indicates where the seed originated. Each seed listed in the table was germinated and each of these were grown into seedlings and the leaves were used for this current microbiome study (five individual trees from Texas and five individual trees from Georgia). We have added a supplemental picture (Figure S2) showing a few of the seedlings from the controlled crosses (from different mother trees).
Reviewer 2 Report
Dear Editors,
I was curious to review a manuscript entitled "Influence of Geographical Orchard Location on the Microbiome from the Progeny of a Pecan Controlled Cross" by Kimberly Cervantes, Ciro Velasco-Cruz, L.J. Grauke, Xinwang Wang, Patrick Conner, Lenny Wells, Clive H. Bock, Cristina Pisani and Jennifer J. Randall.
According to the Authors, the aim of the research presented in this manuscript was to gain insight into the influence that maternal tree origin has on the composition of the microbiota associated with developing pecan seedlings.
The research results provide interesting information on the possibility of increasing plant productivity. The research seems to be extremely original and innovative. I believe that the research was properly conducted and the manuscript is correctly prepared. However, I have a few comments that will help the authors refine this manuscript:
1) Keywords - should be different from the words used in the title of the manuscript, and moreover, the keywords should be arranged in alphabetical order.
2) Abstract - please include here a concisely formulated hypothesis and underline the practical significance of the results obtained.
3) Figures 1 - please make the font bigger on the X and Y axes and in the legend. GA and TX should be placed horizontally.
4) Figures 2 - please make the font bigger on the X and Y axes. GA and TX should be placed horizontally.
5) Figure 3 - please make the font larger on the X and Y axes and in the legend.
6) Table 1 - please explain the abbreviations GA and TX. Please note that all tables and figures should be understood without having to look for explanations of abbreviations in the text of the manuscript.
7) I did not find the Conclusions chapter in the manuscript. Although it is not mandatory, I highly recommend adding this chapter.
8) Please adapt the entire manuscript with greater care to the requirements of the template binding in the journal Plants. Please pay particular attention to the References section.
In conclusion, I believe that due to the interesting subject matter of the research presented in this manuscript, the Editors of the journal Plants should consider publishing this manuscript.
Author Response
Thank you for your time and effort reviewing this manuscript. We have addressed your concerns and comments please see our response below.
“The research results provide interesting information on the possibility of increasing plant productivity. The research seems to be extremely original and innovative. I believe that the research was properly conducted and the manuscript is correctly prepared. However, I have a few comments that will help the authors refine this manuscript:”
1) Keywords - should be different from the words used in the title of the manuscript, and moreover, the keywords should be arranged in alphabetical order.
Response: DONE. Please see line 29.
2) Abstract - please include here a concisely formulated hypothesis and underline the practical significance of the results obtained.
Response: Done. The hypothesis is underlined, and the practical significance of the results were added.
3) Figures 1 - please make the font bigger on the X and Y axes and in the legend. GA and TX should be placed horizontally.
Response: The font sizes were made larger for the x and y axes titles. We ensured that the y axis did meet the font guidelines in Plants.
4) Figures 2 - please make the font bigger on the X and Y axes. GA and TX should be placed horizontally.
Response: The font sizes were made larger for the x and y axes titles. We ensured that the y axis did meet the font guidelines in Plants.
5) Figure 3 - please make the font larger on the X and Y axes and in the legend.
Response: Done.
6) Table 1 - please explain the abbreviations GA and TX. Please note that all tables and figures should be understood without having to look for explanations of abbreviations in the text of the manuscript.
Response: Thank you for the suggestion. This was modified as requested.
7) I did not find the Conclusions chapter in the manuscript. Although it is not mandatory, I highly recommend adding this chapter.
Response: Thank you for your suggestion a conclusion section was added (lines 414-429).
8) Please adapt the entire manuscript with greater care to the requirements of the template binding in the journal Plants. Please pay particular attention to the References section.
Response: Done. We adjusted the text to the template. We have adjusted the references to the correct format.
Reviewer 3 Report
Dear author(s),
The following points should clearly be corrected and explained for readers:
Abstract
1. L14, scientific name of pecan should be written with author name such as Carya illinoinensis (Wangenh.) K.Koch in the first mentioned place.
2. The main findings should be given in abstract.
Keywords
3. L29, Remove numbers. Scientific name could be written as Carya illinoinensis. Are site or location names important enough to be put into keywords?
Introduction
4. L32, author name of Carya illinoinensis such as (Wangenh.) K.Koch. should be written.
5. L61, write author name of disease such as Venturia effusa (G. Winter) Rossman & W.C. Allen
6. L67-83, this para including climatic data should be given as findings in RESULTS with a subtitle. You can draw a Figure, too.
7. L68, check sentence? An active sentence may be better.
8. L71, precipitation should be given as mm.
9. L78, precipitation should be given as mm.
10. L85-89, author name of these diseases should be written.
11. L89-90, scientific name and author name of these diseases should be written.
12. L103, author name of plant species should be written.
Results
13. L120, first full name as operational taxonomic units (OUT) and then abbreviation for OUT.
14. L124, please check 264 OTUs. 128 + 217 = 345, 345-61 = 284. Is it true? Or explain please?
15. L129, check 31 fungal OTUs again? 22 + 18 = 40, 40 – 8 = 32. Explain it for readers?
16. L120-132, Of the 264 OTUs aggregated by taxonomic family in L124 and Of the 31 fungal OTUs aggregated by taxonomic family. Which one belong to bacteria?
17. L134-137, if you know author names of these bacterial families please write them.
18. L145, please explain EC and then use its abbreviation.
19. L189, change PCoA to PCA. PCA is common
Discussion
20. L201-202, please cite a references about “… fungicide treatments.”
21. L212, delete gamma before “…, are…”.
22. L216, author name of wheat should be written as L.
23. L217, author name of common bean should be written as L.
24. L217, author name of Arabidopsis thaliana should be written as (L.) Heynh.
25. L237-238, author names of diseases should be written in the first mentioned place.
26. L255-256, these findings could be written in abstract.
27. L257, scientific and author name of cottonwood tree should be written.
28. L285, cite a relevant reference after “…; as reported in the literature,”…
29. L292, scientific and author name of soybean should be written.
30. L309-311, this sentence could be given in a conclusion and the sentence could be written in abstract, too.
M&M
31. L334, change “… Seed Information” to “… seed information.”
32. In Table 1, the words in the last six columns could be shortened as:
|
Nut |
|||||
|
Length (mm) |
Width (mm) |
Height (mm) |
Weight (g) |
Volume (?) |
Density (?) |
|
|
|
|
|
|
|
33. L338, please write a subtitle for traits such as Studied Traits. Volume Density should be explained. Write these findings for traits in RESULTS for reaaders.
34. L348, you have done more than one analysis. Thus, change “… analysis” to “… analyses”
35. L384, delete (PCo) and write “… analysis (PCA),
I have read your mn with great pleasure.
Author Response
Thank you for your time and effort reviewing this manuscript. We have addressed your concerns and comments please see our response below.
Abstract
- L14, scientific name of pecan should be written with author name such as Carya illinoinensis () K.Kochin the first mentioned place.
Response: Done
- The main findings should be given in abstract.
Response: Done.
- L29, Remove numbers. Scientific name could be written as Carya illinoinensis. Are site or location names important enough to be put into keywords?
Response: Done. Keywords were adjusted.
Introduction
- L32, author name of Carya illinoinensissuch as () K.Koch. should be written.
Response: Done.
- L61, write author name of disease such as Venturia effusa(G. Winter) Rossman & W.C. Allen
Response: Done.
- L67-83, this para including climatic data should be given as findings in RESULTS with a subtitle. You can draw a Figure, too.
Response: The climatic information that is included in the introduction were taken from literature and these measures were not measured by the authors. Therefore, the information introducing the reader to differences between Georgia and Texas will remain in the introduction.
- L68, check sentence? An active sentence may be better.
Response: This sentence was altered. See lines L68-69.
- L71, precipitation should be given as mm.
Response: Done.
- L78, precipitation should be given as mm.
Response: Done.
- L85-89, author name of these diseasesshould be written.
Response: Done.
- L89-90, scientific name and author name of these diseasesshould be written.
Response: Done.
- L103, author name of plant species should be written.
Response: Done.
Results
- L120, first full name as operational taxonomic units (OUT) and then abbreviation for OUT.
Response: Done.
- L124, please check 264 OTUs. 128 + 217 = 345, 345-61 = 284. Is it true? Or explain please?
Response: The text has been altered to make it clearer. There was a total of 264 OTUs aggregated at the family level. Of these, 81 OTUs aggregated at the family level were identified in both states, as mentioned in L136-139. (The ‘61’ was a typo and meant to be 81.) In the Texas seedlings, 47 unique OTUs aggregated by taxonomic family were identified, while 136 were identified in the Georgia seedlings.
81 + 47 + 136 = 264
- L129, check 31 fungal OTUs again? 22 + 18 = 40, 40 – 8 = 32. Explain it for readers?
Response: This has been clarified for the readers. L142-145. (The ‘31’ was a typo and meant to be 30.)
10 + 12 + 8 = 30
- L120-132, Of the 264 OTUs aggregated by taxonomic family in L124 and Of the 31 fungal OTUs aggregated by taxonomic family. Which one belong to bacteria?
Response: The 264 OTUs aggregated at the family level belong to the bacterial communities. This has been clarified in the text. L136-139 and L142-145.
- L134-137, if you know author names of these bacterial families please write them.
Response: Done.
- L145, please explain EC and then use its abbreviation.
Response: Done.
- L189, change PCoA to PCA. PCA is common
Response: Line 189 (now L205) is PCoA not a PCA. This was not changed.
Discussion
- L201-202, please cite a references about “… fungicide treatments.”
Response: There is not a reference for the GA orchard fungicide treatment the orchard treatments are discussed in Materials and Methods.
- L212, delete gamma before “…, are…”.
Response: Done.
- L216, author name of wheat should be written as L.
Response: Done.
- L217, author name of common bean should be written as L.
Response: Done.
- L217, author name of Arabidopsis thaliana should be written as () Heynh.
Response: Done.
- L237-238, author names of diseases should be written in the first mentioned place.
Response: Done.
- L255-256, these findings could be written in abstract.
Response: Done.
- L257, scientific and author name of cottonwood tree should be written.
Response: Done.
- L285, cite a relevant reference after “…; as reported in the literature,”…
Response: Done.
- L292, scientific and author name of soybean should be written.+
Response: Done.
- L309-311, this sentence could be given in a conclusion and the sentence could be written in abstract, too.
Response: Thank you for the suggestion. A form of this sentence is now in the abstract and in the conclusion section (lines 414-427).
M&M
- L334, change “… Seed Information” to “… seed information.”
Response: Change was made.
- In Table 1, the words in the last six columns could be shortened as:.
Response: Edits were done as requested.
- L338, please write a subtitle for traits such as Studied Traits. Volume Density should be explained. Write these findings for traits in RESULTS for reaaders.
Response. A new section was added to the results and Table 1 was added to this section (L120-130)
- L348, you have done more than one analysis. Thus, change “… analysis” to “… analyses”
Response. Done.
- L384, delete (PCo) and write “… analysis (PCA),
Response. This line actually refers to the PCoA plot so this will not be changed.
Round 2
Reviewer 1 Report
Manuscript "Influence of Geographical Orchard Location on the Microbiome from the Progeny of a Pecan Controlled Cross"
by Kimberly Cervantes, Ciro Velasco-Cruz, LJ Grauke, Xinwang Wang, Patrick Conner, Lenny Wells, Clive H Bock , Cristina Pisani, Jennifer J Randall The necessary corrections were taken into account and made to the manuscript. This material may be accepted for publication in its present form.